# Beyond Purification: Evolving Roles of Fusion Tags in Biotechnology

**DOI:** 10.3390/cimb47090768

**Published:** 2025-09-17

**Authors:** Tsutomu Arakawa, Teruo Akuta

**Affiliations:** 1Alliance Protein Laboratories, 13380 Pantera Road, San Diego, CA 92130, USA; 2Research and Development Division, Kyokuto Pharmaceutical Industrial Co., Ltd., 3333-26, Aza-Asayama, Kamitezuna, Tahahagi-shi 318-0004, Ibaraki, Japan; t.akuta@kyokutoseiyaku.co.jp

**Keywords:** fusion tag, soluble expression, insoluble expression, pull-down, tag removal, intein, purification, half-life

## Abstract

Genetic fusion of a tag sequence to a target protein, or protein of interest (POI), is one of the most widely used technologies for recombinant expression. Tag-fusion proteins can enhance soluble expression, prolong half-life, increase binding avidity, and facilitate protein purification or refolding. In addition, tag-fusion proteins can be used to identify POI-binding partners through pull-down or immunoprecipitation assays. Beyond these classical applications, tags have evolved to serve as multifunctional tools, enabling real-time imaging, spatial localization, targeted delivery, and regulation of protein activity in living systems. Some engineered tags also allow conditional control, such as pH or ligand-dependent stabilization, thus expanding their utility in synthetic biology and therapeutic design. Here, we summarize protein-based and peptide-based tags, as well as methods for tag removal. While not fully comprehensive, this review aims to help researchers design suitable tag formats for specific goals.

## 1. Introduction

Techniques for efficient expression of native recombinant proteins have become increasingly important with recent advances in genomic and proteomic studies [1,2,3] and therapeutic applications [4,5,6,7]. The most conventional recombinant *Escherichia coli* (*E. coli*) expression system often leads to insoluble heterologous proteins, a major bottleneck for the current systematic studies of protein characterization and production [8,9,10,11,12]. When expressed as inclusion bodies, developing less time-consuming refolding technologies is required, which often involves trial-and-error approaches [13]. Soluble expression simplifies recombinant protein production. One of the approaches to enhance soluble expression is co-expression of protein chaperones [14]. Even machine learning is used to predict the approach for soluble expression [15].

Protein fusion techniques are a powerful method for the expression of heterologous proteins in *E. coli* [16,17,18]. Fusion proteins can be readily identified and purified using the unique affinity of the fusion tags, e.g., antibody or ligand to the tag proteins/peptides. Furthermore, the fusion tags may increase the solubility and folding efficiency of target proteins in fusion constructs, which eliminates troublesome refolding experiments [19,20]. In this review, we summarize various protein- or peptide-based fusion tags and the advantages of their applications.

## 2. Protein Tag

Many protein- or peptide-based fusion tags have been developed, and commonly-used tags are described below, although there may be more less-commonly used tags. Figure 1 shows a diagram of fusion tag formats. Number 1 is a protein tag (blue) fused to a target protein or protein of interest (POI, red); number 2 is a protein tag fused to a target peptide; and number 3 is a short peptide tag fused to a target protein. The tags are often intervened by a linker peptide (green), which may or may not be cleavable. Some properties and utility of the protein-based tags as well as pros (advantages) and cons (limitations) are summarized in Table 1.

### 2.1. Green Fluorescent Protein (GFP)

GFP is a small (molecular weight of 27,000–30,000) acidic protein with the isoelectric point of around 5.6 and has been used as a reporter of expression, location, and interaction of various intra-cellular or extra-cellular proteins [21,22,23]. When GFP is fused to a protein of interest (POI) or “target protein” at the N-terminus as GFP-POI or at the C-terminus as POI-GFP and expressed in either eukaryotic or prokaryotic cells, the fluorescence of GFP is normally considered to indicate expression of the POI and its location [24,25]. This assumes that both the GFP and POI spontaneously fold independently from each other and are functional. Such folding behavior of at least the GFP moiety may be supported by the highly reversible nature of its denaturation and renaturation processes around neutral pH, demonstrated by recovery of fluorescence upon various denaturation mechanisms [26]. However, note that although it is true that complete denaturation abolishes GFP fluorescence, recovery of fluorescence does not perfectly parallel the structure recovery, meaning that even a partial refolding could lead to fluorescence recovery, as demonstrated by urea denaturation at pH 6.2 slightly above the pI of GFP (pH 5.6), which does not appear to lead to full denaturation [27]. In fact, GFP is extremely stable against various denaturation mechanisms and also readily renatures, especially above neutral pH [28]. This assures a likely possibility that the GFP moiety folds regardless of the fused target protein. Epidermal growth factor is a small protein containing three disulfide bonds and has been produced using many different expression systems [29,30,31,32]. Its expression can be readily observed by fluorescence of GFP.

GFP was used to increase soluble expression of recombinant proteins [33]. GFP was fused to a pea actin isoform (PEAc1) with a purification His-tag to increase the solubility in *E. coli* expression. The C-terminus of PEAc1 was fused to the N-terminus of GFP as a PEAc1-GFP and expressed in the cytoplasm of *E. coli* [34]. The GFP fusion resulted in higher soluble expression of the PEAc1 actin in the native structure and, thus, can polymerize normally to F-actin [35]. Direct expression of folded actin has not been successful in *E. coli* due to the absence of appropriate molecular chaperons in *E. coli*. Refolding of insoluble actin is also rather difficult [36,37]. Insoluble SARS-CoV-2 RNA polymerase, when expressed in *E. coli*, was made soluble by GFP fusion, and even further in expression with a dual fusion with SUMO [38].

### 2.2. Thioredoxin (Trx)

Trx is a small redox-related protein and has been used as a fusion tag [39,40] or a protein chaperone [41,42,43,44] to increase soluble expression. This redox function was proposed to play a role in increasing recombinant expression of human proteins in *E. coli* [41]. The above study [41] expressed eight human proteins, including oncogenes, kinases and transcription factors, important drug targets in *E. coli*, largely as insoluble IBs. When either molecular chaperone or Trx or both was co-expressed, only Trx co-expression improved soluble expression for all eight proteins, which was ascribed to a more reduced environment of *E. coli* cytoplasm by expressed Trx. A similar effect of Trx co-expression was also observed for other proteins [42]

Trx is also useful as a fusion tag. For example, bromelain from a pineapple, which is not soluble when expressed by itself in *E. coli*, was made soluble by fusing to Trx tag [39]. A simple generation of a cytokine, interleukin-24, was made possible using Trx-fusion expression [40]. Trx was fused to the N-terminus of the SARS-CoV-2 Nucleocapsid protein (NP) to enhance its solubility and stability [45]. The acidic property of Trx fusion tag resulted in efficient binding to an anion-exchange resin and removal of nucleic acids that can bind to the NP [45]. When applied to membrane proteins, e.g., human cystic fibrosis transmembrane conductance regulator and a bacterial ATP synthase, Trx tag offered more effective purification after heat treatment [46].

**Table 1 cimb-47-00768-t001:** Properties and applications of protein fusion tags.

Protein Tag	Size	Main Uses	Oligomerization	Advantages (Pros)	Limitations (Cons)	Ref.
GFP	27 kDa	Detection, solubility	N	Direct fluorescence monitoring; stabilizes fusion proteins	Moderate (~27 kDa); may affect folding/function	[21,22,23,24,25]
Trx	12 kDa	Solubility, folding	N	Enhances folding in *E. coli*; improves solubility	Limited purification use; may require removal	[39,40,45]
MBP	42.5 kDa	Solubility, purification	N	Strong solubility enhancer; affinity purification on amylose resin	Large (~42 kDa); may alter activity	[47,48,49,50,51,52,53,54,55]
NusA	55 kDa	Solubility	N	Very strong solubility enhancer for insoluble proteins	Very large (~55 kDa); usually removed	[56,57,58,59]
HSA	66 kDa	Stability, half-life extension	N	Extends serum half-life; enhances solubility; clinically validated	Large (~66 kDa); may interfere with activity	[60,61,62,63,64,65]
SUMO	11 kDa	Solubility, cleavage	N	Enhances folding/solubility; precise cleavage by SUMO protease	Requires SUMO protease; adds extra step	[66,67,68,69,70,71,72]
BLA	29 kDa	Solubility, stability	N	Highly stable under extreme/halophilic conditions	Not widely adopted; limited affinity purification	[73,74,75,76,77]
GST	26 kDa (mono) 52 kDa (dimer)	Purification, solubility, IP	Dimer	Affinity purification with glutathione resin; moderate solubility enhancer	Dimerization (26 kDa × 2); may alter activity and lead to false-positive in IP	[78,79,80,81,82,83,84,85,86,87,88]
Fc	25 kDa (mono), 50 kDa (dimer)	Purification, stability	Dimer	Protein A/G affinity purification; increases stability and half-life, Fc receptor binding	Large (~25 kDa/chain); promotes dimerization	[89,90,91,92,93,94,95,96,97]
C-prpeptide of α1 collagen	34 kDa	Folding, stability	Trimer	Enhanced activity (e.g., TRAIL)	Large domain; mainly suited for limited target	[98,99]
Streptavidin	13 kDa (mono), 52 kDa (tetramer)	Immobilization, purification	Tetramer	Extremely high biotin affinity; useful for fusion constructs (e.g., VHH-Streptavidin for column binding)	Tetramerization may affect fusion partner; very strong binding can hinder elution	[100,101,102]

### 2.3. Maltose-Binding Protein (MBP)

Maltose-binding protein (MBP) is a relatively large protein among many fusion proteins, with a molecular weight of ~42 kD. It is able to enhance the solubility of target proteins and can be readily purified by amylose resin [47]. High expression of soluble, active tobacco etch virus protease in *E. coli* was achieved by fusing it to the C-terminus of *E. coli* MBP [48]. Such high soluble expression by MBP fusion was observed as a fusion to inhibitory protein against snake venom [49]. MBP as a fusion appears to commonly improve the solubility of target proteins [50,51,52]. For example, recombinant prolactin, produced in *E. coli* as an MBP fusion, resulted in enhanced soluble expression of stable, functional protein [53]. High soluble expression of human oncostatin M was achieved with MBP fusion [54]. A viral ATPase 2C fused with MBP showed high solubility as a membrane-bound state, indicating that MBP can lead to enhanced expression of lipid–protein complex [55]. High soluble expression of a viral capsid protein VP2 vaccine in *E. coli* was achieved with MBP fusion, among various fusion tags [52].

### 2.4. NusA

NusA is is N-Utilization Substance A, which can improve the solubility of the target proteins [56]. Astaxanthin, a marine carotenoid used in various applications, binds to a protein called crustacyanin that can solubilize the astaxanthin. The recombinant crustacyanin of a lobster, which was found in IBs when expressed in *E. coli*, was fused to NusA-tag for soluble expression, resulting in soluble protein binding to and solubilization of astaxanthin [57]. The N-terminal domain of human protein phosphatase 1 nuclear targeting subunit achieved soluble expression in *E. coli* as a fusion to NusA [58]. A functional lysyl oxidase was expressed as a fusion to NusA [59].

### 2.5. Human Serum Albumin (HSA)

Serum albumin has long half-life, is stable against proteolytic degradation, and has few immunological reactions [60]. HSA is a simple protein (non-glycosylated) abundant in human plasma and has a relatively long half-life of ~14 days. The non-glycosylate property of HSA makes it possible to produce it in prokaryotic bacteria, although glycosylation can extend the half-life of proteins in serum. HSA is well known to bind various organic compounds, e.g., as a carrier of drug substances. The usefulness of HSA as a fusion tag has been well summarized [61]. The HSA-CD4 fusion showed inhibitory activity on AIDS virus entry into CD4+ cells, more effectively than CD4 by itself [60]. The pharmacokinetic properties of interferon-β were greatly improved through fusion to HSA [62]. Anti-oxidant activity of thioredoxin (Trx) was used to treat respiratory diseases with a fusion to HSA [63].

HSA fusion with human lactoferrin improved the half-life and efficacy for inhibiting cancer cell migration through its effects on metalloproteinase [64]. Half-life extension by HSA was achieved for short half-life small peptide and glucagon-like peptide 1 [65].

### 2.6. SUMO

SUMO (Small Ubiquitin-like Modifier) tag fusion, derived from SUMO protease, has been shown to be effective in soluble expression [66]. SUMO was found to be effective in increasing soluble expression of many proteins, including SARS-CoV-2 [38,67], spider dragline silk protein [68], a venom protein JarC [69], apoptin, and a tumor-cell-specific apoptosis inducer [70]. Human interleukin-7, a therapeutically important cytokine, was expressed as a soluble form by fusing to SUMO tag in *E. coli*, which showed IB formation when expressed alone [71]. In particular, SUMO was found to be effective in enhancing soluble expression of difficult-to-express proteins [72].

### 2.7. Halophilic β-Lactamase (BLA)

Halophilic proteins, such as halophilic β-lactamase (hBLA) produced by halophilic archaea and bacteria, show distinct characteristics in their primary structure of being highly acidic [73]. This acidic property and extensive negative charges at neutral pH makes the unfolding/refolding highly reversible [74,75]. The hBLA by itself was efficiently expressed in *E. coli* cytoplasm as a fully folded active form, suggesting that it may be effective as a fusion tag.

Human interleukin 1α (IL1α) is a typical aggregation-prone protein in the *E. coli* expression system [76]. When His-IL1α (his-tag IL1α) was expressed in *E. coli*, it was observed entirely in the pellet [77]. On the contrary, expression with BLA tag (His-BLA IL1α fusion protein) largely resulted in soluble fraction. Another aggregation-prone protein, human serine racemase (hSR), formed inclusion bodies, while as a BLA-fusion protein, most of the fusion protein appeared in the soluble fraction. In another example, recombinant expression of a halophilic α-amylase in *E. coli* cytoplasm with N-terminal His-tag (His-Amy) resulted in low yield. However, when the protein was expressed as a BLA fusion protein, a much larger amount of the product was detected in the soluble fraction of *E. coli.* Thus, it is evident that expression of “difficult-to-express” proteins as a fusion protein with highly soluble and highly structure-reversible halophilic His-BLA clearly improved the solubility and yield of these three target proteins.

HNP-1, a human neutrophil α-defensin, is a microcidal peptide consisting of 30 amino acid residues and contains 6 Cys residues, which forms three intra-chain disulfide bonds. When expressed with N-terminal His tag, yield of His-tag HNP-1 was low. On the contrary, for His-BLA-HNP1 fusion protein, almost all was detected in the soluble fraction, indicating enhanced solubility that assisted correct formation of disulfide bonds.

### 2.8. Dimeric Glutathione-S-Transferase (GST)

The GST tag is a monomeric ~26 kDa protein that forms a homodimer, as schematically depicted in Figure 2 [78,79], and a classical fusion format for soluble expression and glutathione-affinity purification or pull-down assay of recombinant protein [80,81,82]. Leukemia inhibitory factor was expressed with GST-fusion in *E. coli* as a soluble and functional state [83]. GST fusion was used to generate convenient Grb7-SH2 domains (GST-SH2) to coat BIAcore chip surface and screen-binding peptides [84]. GST tag has been particularly popular for pull-down or immunoprecipitation assay [85,86,87,88]. As depicted in Figure 2, bivalent anti-GST antibody can cross-link GST-fusion POI, to which binds the ligand (POI’s binder). Since GST is a dimer, this antibody cross-linking of GST can cause formation of the multimeric GST-fusion/ligand complex. On the contrary, if the fusion protein is monomeric, antibody binding to the monomeric tag can cause formation of only a dimer. Furthermore, using multivalent anti-tag beads, e.g., glutathione-conjugated beads, can theoretically bind more than two fusion proteins and result in large tag-POI/ligand complexes and effective pull-down results.

### 2.9. Dimeric Fc

The Fc region of human antibody is used as a dimeric fusion tag. When a protein or peptide POI is fused to the Fc, the fusion proteins are called “peptibodies”, as opposed to antibodies, as depicted in Figure 3 and described later [89,90]. The dimeric Fc fusion can carry two target proteins, as shown in Figure 3, and may bind two monomeric ligands, to which the target molecule binds. If the ligand is dimeric, the dimeric fusion may bind to the dimeric ligand bivalently, conferring binding avidity: namely, binding strength is enhanced. On the same token, if the ligand is on the cell surface and the Fc fusion can bind two cell-surface ligands simultaneously, then binding avidity can occur as shown in Figure 3.

One of the most-well known peptibodies or Fc fusions is etanercept, in which the C-terminus of tumor necrosis factor (TNF) receptor extracellular domain is fused to the N-terminus of human Fc domain, as depicted in Figure 4, and binds to TNF, thereby inhibiting inflammatory activities of TNF. TNF is a trimeric ligand, while Fc-fusion biologicals (peptibodies) are dimeric, leading to potential complex binding modes, as shown in Figure 4, where the etanercept binds one TNF molecule with avidity, only one TNF or two TNF molecules, although it is not clear which binding model is more likely for Fc-receptor fusion binding to the trimeric TNF.

There are a number of Fc-fusion applications, as known as peptibodies (Figure 5) [90,91]. Thrombopoietin (TPO) plays a key role in generation of platelets, but has met with limited success, while Fc-fusion peptibody with a short TPO receptor-binding peptide resulted in therapeutic efficacy [92,93,94]. Similarly, an Fc-fusion molecule was generated using a fibroblast growth factor (FGF) receptor-binding peptide and developed as a long-circulating drug carrier by the FGF receptor-mediated internalization of the Fc-fusion molecule, as shown in Figure 5 (right panel) [95]. Another growth factor, epidermal growth factor (EGF), is also involved in cancer, and thus a peptibody-targeting EGF receptor was developed for cancer treatment [96]. An ion channel was targeted by Fc-fusion molecules using a 21-amino-acid channel-binding peptide [97].

### 2.10. Trimeric Human C-Propeptide of α1 Collagen

The self-assembly properties of protein tags are actively being harnessed in the development of novel therapeutic agents. For instance, SCB-313, a fully human TNF-related apoptosis-inducing ligand (TRAIL)–trimer fusion protein, was created by fusing the human C-propeptide of α1 collagen, a trimer tag, in-frame to the C-terminus of human TRAIL, yielding a disulfide bond-linked homotrimer, as shown in Figure 6 [98]. Preclinical studies have demonstrated greater bioactivity, an improved safety profile, and superior pharmacokinetics and pharmacodynamic effects compared with those of the TRAIL–Fc dimer, anti-death receptor 5 agonist antibodies, and related agents [99]. Clinical trials are currently in progress for peritoneal malignancies, malignant pleural effusions, malignant ascites, and peritoneal carcinomatosis, and SCB-313 is expected to play a pivotal role in future cancer therapies [99].

### 2.11. Tetrameric Streptavidin

Streptavidin forms a stable tetramer [100] and, hence, can potentially carry four target proteins when fused to streptavidin, as depicted in Figure 6 [101]. SLG (sodium lauroyl glutamate) was used to solubilize and refold a fusion construct of single-chain Fab fragment (scFv) with streptavidin to create a mulimeric scFv [102]. The fusion protein was expressed in *E. coli* as insoluble inclusion bodies and solubilized by SLG and refolded, resulting in tetramer formation, as depicted in Figure 6.

## 3. Peptide Tags

Similar to protein tags, peptide tags are widely employed in protein research and the biotechnology industry. Their key properties and applications are summarized in Table 2.

### 3.1. His

His-tag is made of multiple histidine residues, mostly six residues, which binds to Immobilized Metal Affinity Chromatography (IMAC) resin in one of the most frequently used types of chromatography. Advantages of IMAC chromatography are its ease of elution by simple competitive binding by imidazole and tolerant to solvent conditions, e.g., binding in the presence of GdnHCl, salt, and arginine, which are often present in purification processes. The ability to bind in the presence of GdnHCl allows direct loading of inclusion bodies solubilized by GdnHCl and on column refolding. As shown in Figure 7, unfolded protein with a His tag, e.g., solubilized inclusion bodies by GdnHCl, can easily bind to IMAC (here immobilized Ni ions) resin and be eluted by imidazole. Alternatively, GdnHCl is removed to induce refolding of the bound protein with His-tag on. The bound and refolded protein is then eluted by either imidazole or proteolytic cleavage of the linker that connects His-tag and the target protein that has been refolded.

His-tag does not necessarily enhance the solubility of expressed proteins, but is widely used for easy application for protein purification. His-tag is, therefore, often combined with protein tags to enhance soluble expression described above [82,103,104]. Such combinations can lead to high soluble expression and easy purification. When purifying soluble fraction, many host–cell proteins that have affinity for metal ions, e.g., histidine-containing proteins can bind to IMAC resin, requiring imidazole concentration adjustment to selectively elute the His-tag proteins, which normally have higher metal affinity.

### 3.2. Flag

Flag tag is an eight-amino-acid-long (DYDDDK) peptide, which has been extensively used to facilitate affinity purification. Elution of the bound proteins is carried out through competition with the flag peptide, which has high affinity and specificity to anti-flag antibody [105]. Affinity of flag tag for the antibody was further enhanced by triple-flag sequence fused to the target proteins [106,107]. Flag tag is widely used for immunodetection because of the large availability of monoclonal antibodies and compatibility with various experimental systems [108]. It is particularly effective in protein localization studies by immunofluorescence, as the small size of the tag minimizes steric hindrance. Furthermore, flag-tag fusion has been applied in structural studies, enabling precise monitoring of protein–protein interactions and complex formation.

### 3.3. Myc

Myc fusion tag is derived from an antibody epitope (EQKLISEEDL) of human c-Myc protooncogene and exhibits high specificity for a commercial antibody; thus, it can be used for one-step purification of proteins. The Myc tag was used to detect expression of SARS-CoV-2 fused to the tag and serves as a membrane-directing signal peptide [109]. Myc tag fused to single chain antibody (scFv) was shown to possess adverse effects on CD19 CAR-T therapy by anti-CD19 scFv, despite stabilization of the scFv by Myc tag [110].

Myc tag is especially common in cell biology for Western blotting, immunoprecipitation, and immunofluorescence studies [111]. Its well-characterized antibody recognition provides robust detection across diverse experimental systems. However, care is needed in therapeutic applications because the tag can influence protein folding or immune responses. It remains a powerful option for analytical assays where high specificity is required.

### 3.4. PA

PA tag is a 12-amino-acid peptide (sequence: GVAMPGAEDDVV) commonly used for protein detection and purification [112]. It is specifically recognized by the high-affinity anti-PA tag monoclonal antibody clone NZ-1. Recombinant proteins fused to the PA tag can be effectively purified using antibody-immobilized affinity columns. The PA tag provides a highly selective and stable interaction with its antibody, which has been exploited in structural biology and crystallography [112]. Recombinant proteins bound to anti-PA antibody columns can be efficiently eluted not only by competitive peptide or acidic buffers but also under near-neutral conditions using 10 mM MES- 2 M MgCl_2_ (pH 6.0) [113]. Remarkably, this MgCl_2_-based elution does not cause protein denaturation, allowing recovery of recombinant proteins in their native states. This unique property enables purification of highly pure and functionally active proteins under mild conditions, which is particularly advantageous for sensitive proteins and downstream functional studies.

### 3.5. HA

HA tag is a short fusion tag derived from hemagglutinin (HA) and contains nine amino acids, YPYDVPDYA. HA-tag fusion was used to monitor the function of mitofusion protein 1 and 2 (MFN1 and MFN2) to cause organelle membrane fusion activities of these proteins [114]. HA tag was used to construct more specific binding proteins for protein–protein interaction analysis by combining with other tags [115]. As a result of its small size and highly specific antibody recognition, the HA tag is widely used in protein localization and trafficking studies, especially with microscopy-based methods [116]. It is also applied in immunoprecipitation, protein complex isolation, and reporter assays [117]. The broad commercial availability of anti-HA antibodies ensures reproducibility across laboratories and facilitates multipurpose usage.

### 3.6. V5

A V5 tag is a 14-amino-acid peptide, GKPIPNPLLGLDST, derived from the C-terminus of the human paramyxovirus. The extracellular domain of interleukin-5 and granulocyte macrophage-colony-stimulating factor was fused to V5 fusion tag and purified for the analysis of their interactions with each protein counterpart [118]. The V5 tag was used to express the SARS-CoV M gene fragment was expressed as a fusion to V5 tag in Vero cells for the analysis of protein expression [119]. V5-tagged nanobody was developed to detect intra-cellular binding proteins [120].

The V5 tag is especially useful in immunodetection assays because it is less commonly found in natural mammalian proteins, minimizing background signals [118]. It supports a variety of applications, including protein expression analysis, interaction studies, and nanobody-based detection. Its relatively larger size compared to HA or Myc tag may occasionally affect folding, but in most cases it offers robust functionality with high antibody specificity.

**Table 2 cimb-47-00768-t002:** Properties and applications of peptide tags.

Protein Tag	Sequence	Size (aa)	Main Uses	Ligand/ Antibody	Advantages (Pros)	Limitations (Cons)	Ref.
His	HHHHHH	6.8	Affinity purification + detection	Ni^2+^/Co^2+^-NTA resin, anti-His antibody	Highly standardized purification Can bind in the presence of denaturant	Non-specific binding, requires metal ions, can co-purify contaminants	[82,103,104]
Flag	DYKDDDDK	8	Affinity purification + detection	Anti-FLAG antibody, FLAG-beads	Small size, efficient immunoprecipitation and elution	Acidic elution may affect proteins	[105,106,107,108]
PA	GVAMPGAEDDVV	12	Affinity purification + detection	NZ-1 antibody	High-affinity antibody binding, efficient IP and purification	Less commonly used than FLAG/HA/Myc	[112,113]
Strept tag II	WSHPQFEK	8	Affinity purification + detection	Strep-Tactin resin	Mild elution, biotin-based binding, good for native proteins	Lower affinity than Twin-Strep	[121,122,123,124]
Twin-Strept tag	WSHPQFEK × 2 (linked)	16	High-efficiency purification + detection	Strep-Tactin XT resin	Much higher affinity than single Strep-tag, works well in mammalian systems	Larger size than single tags	[121,122,123,124]
Myc	EQKLISEEDL	10	Detection (WB, IF, IP)	Anti-Myc antibody	Widely used, multiple antibodies available	Poor for purification	[109,110,111]
HA	YPYDVPDYA	9	Detection (WB, IF, IP)	Anti-HA antibody	Excellent for immunostaining and localization	Not efficient for purification	[114,115,116,117]
VA (V5)	GKPIPNPLLGLDST	14	Detection (WB, IF)	Anti-V5 antibody	Affinity purification with glutathione resin; moderate solubility enhancer	Dimerization (26 kDa × 2) may alter activity and lead to false-positive in IP	[118,119,120]
Dimerization	EFLIVIKS	7	Dimerization	Not available	Protein A/G affinity purification; increases stability and half-life, Fc receptor binding	Large (~25 kDa/chain); promotes dimerization	[125,126,127]
Insoluble	GIFQINSRY GILQINSRW	9.10	Purification by precipitation	Not available	Enhanced activity (e.g., TRAIL)	Large domain; mainly suited for limited targets Requires solubilization for application	[128,129,130,131,132,133,134,135,136,137]

### 3.7. Strep-Tag II or Twin-Strep-Tag

Streptavidin is used as not only a fusion tag (Section 2.11) but also a chromatographic resin, called “Strep-Tactin”, which binds Strep-tag II consisting of an eight-amino-acid peptide, WSHPQFEK, at the biotin binding pocket of Streptavdin (Figure 6). Strep-tag II and the higher-affinity Twin-Strep-tag are widely used for recombinant protein purification [121]. They bind specifically to Strep-Tactin, a proprietary streptavidin variant from IBA, under mild, physiological conditions, enabling recovery of active proteins with minimal contaminants. Elution with desthiobiotin provides gentle release without denaturation. Since these tags are small and minimally disruptive, they can be efficiently fused to the C-terminus of the protein of interest (POI), making them well suited for purifying soluble proteins, membrane proteins, and protein complexes with high purity [122,123].

The Strep-tag system is often preferred for purification of labile or multi-subunit proteins, as the elution process is mild and preserves activity. In particular, the use of desthiobiotin allows elution under near-physiological, neutral pH conditions, thereby minimizing the risk of acid- or chemical-induced protein denaturation [124]. The Twin-Strep-tag provides even higher affinity, making it advantageous for low-abundance or fragile protein complexes [121]. These properties have made the system highly valuable in proteomics, functional assays, and structural studies where preservation of native protein activity is crucial.

### 3.8. Dimerization Tag

EFLIVIKS-fused peptides involve EFLIVIKS or EFLIVKS, the established amino acid sequence to form a homodimer [125,126]. This tag is normally fused directly to the target proteins without linker sequences and has been used to enhance the neuroprotective activity of Humanin [127]. This tag induces stable dimer formation, thereby modulating biological function. It is especially valuable in therapeutic peptide development, where multimerization can increase potency or half-life. Beyond Humanin, the principle has potential in designing synthetic biologics and engineered signaling molecules. However, its application is specialized, as forced dimerization can also alter natural protein function.

### 3.9. Insoluble (Ins)

Insoluble (Ins) is a unique expression tag that results in reduction of POI solubility and thereby its precipitation; it has been identified in the sequence of hen egg lysozyme [128]. Hen egg protein is known as a rich source of many useful peptides for food and technology industries [129]. There have been attempts to generate insoluble but active proteins for simple downstream processing of recombinant proteins [130,131,132,133]. A peptide tag derived from lysozyme demonstrated efficient production of insoluble recombinant native protein and is made of GIFQINSRY derived from human lysozyme and GILQINSRW derived from hen egg lysozyme [134,135,136]. The hen egg insoluble peptide (Ins) was fused to the N-terminus of a highly soluble halophilic Histidine-rich metal binding Protein (HP). The HP contains many histidine residues in series and can absorb metal ions. When HP was expressed in *E. coli*, the protein was largely observed in soluble fraction, reflecting its halophilic nature of high solubility. On the contrary, expression of HP as an Ins-tag fusion, the HP was largely observed in insoluble fraction, demonstrating effectiveness of the Ins-tag to reduce the solubility of highly soluble halophilic protein. The insoluble HP was found to be capable of binding Ni and Zn ions. A similar experiment was carried out for a highly soluble starch-binding protein domain (SBD) of halophilic α-amylase [137], demonstrating soluble expression of SBD and insoluble expression of Ins-tag SBD [128].

## 4. Cleavage of Fusion Tags from Target Proteins

A linker sequence is often incorporated between the target protein (POI) and tag sequence and, in many cases, designed to be cleaved to generate tag-free proteins of interest. After purifying epitope-tagged fusion proteins using affinity resins, cleavage enzymes are used to isolate tag-free, intact proteins of interest [138,139,140]. In this review, we classify tag cleavage methods into three categories. The first involves conventional proteases, the most widely used approach, which relies on protease recognition sequences incorporated into the linker to remove tags but often leaves extra amino acids at the cleavage site. The second category includes methods that enable precise tag removal without leaving any residual amino acids. The third, a rapidly advancing approach, utilizes split intein systems, depicted in Figure 8, to achieve seamless tag removal with high specificity and no additional residues.

### 4.1. Cleavage Enzymes Degrade Proteins Through Cleaving Peptide Bonds

Table 3 summarizes commonly used cleavage enzymes that match the affinity purification scheme based on the sequence they recognize. These proteases include thrombin [141,142,143,144], factor Xa [141,145], enterokinase [146], human rhinovirus (HRV) 3C protease/PreScission Protease [147], and tobacco etch virus (TEV) protease [148]. Extensive information on protease tag cleavage is available in reviews [138,139] and supplier literature, so only a brief summary is included here.

Protease-mediated tag removal enables recovery of proteins close to their native form and is widely applied in structural and functional studies. It offers high sequence specificity, but drawbacks include incomplete cleavage, residual extra residues at the cleavage site, and the cost of enzymes.

Careful optimization is often required to balance efficiency and specificity.

**Table 3 cimb-47-00768-t003:** Conventional cleavage enzymes.

Cleavage Enzyme	Recognized Sequence	Ref.
Thrombin	Leu-Val-Pro-Arg↓Gly-Ser↓	[141,142,143,144]
Factor Xa	Ile-(Glu or Asp)-Gly-Arg↓X	[141,145]
Enterokinase	Asp-Asp-Asp-Asp-Lys↓X	[146]
HRV 3C/PreScission Protease	Leu-Glu-Val-Leu-Phe-Gln↓Gly-Pro	[147]
TEV protease	Glu-Asn-Leu-Tyr-Phe-Gln↓Gly/Ser	[148]

The ‘↓’ symbol indicates the cleavage site.

#### 4.1.1. Thrombin

Thrombin has been widely adopted to remove purification tags (such as His-tags or GST) in recombinant protein workflows by specifically cleaving the LVPR↓GS (“↓” indicates the cleavage site) sequence between the tag and the target protein. A widely cited methodology is presented by Hefti et al. [149], who described a general two-step approach: initial His-tag-based IMAC purification followed by thrombin digestion, yielding native-like protein with only three extra residues at the N-terminus. A broad methodological and mechanistic analysis was provided in a critical review discussing thrombin versus factor Xa cleavage: this review emphasizes both the specificity of thrombin at the consensus site and the risk of off-target cleavage in the fusion partner (target protein or POI), urging careful characterization of the final product [150]. A broader overview [139] describes thrombin among other proteases, noting that while thrombin is active over a broad pH range and compatible with many buffers, it may leave two additional residues (Gly-Ser) at the N-terminus. Together, these studies illustrate thrombin’s advantages in tag removal—such as established specificity and compatibility with detergents—and its limitations, including potential promiscuous cleavage and residual N-terminal extra residues [151,152].

#### 4.1.2. Factor Xa

Factor Xa is a serine protease that specifically cleaves a sequence I(E/D) GR↓X, where X can be any amino acid except arginine or proline, enabling removal of N-terminal affinity tags with minimal residual amino acids at the target protein’s N-terminus [139]. Comprehensive guidance from [140] critically reviews methods using both thrombin and Factor Xa, emphasizing Factor Xa’s capacity for precise cleavage but also noting risks of off-target (within the sequence of target protein) proteolysis—thus urging thorough post-cleavage product characterization. As a disulfide-linked enzyme (17 and 42 kDa), it is sensitive to reducing agents and chelators like EDTA [153,154].

#### 4.1.3. Enterokinase

Enterokinase (enteropeptidase) specifically cleaves after the DDDDK↓X motif (X can be any amino acid but Pro), enabling precise removal of N-terminal tags without residual amino acids. Yuan & Hua [155] demonstrated efficient expression and use of bovine enterokinase light chain as a fusion cleavage enzyme in *E. coli*. A review [140] compared enterokinase with other proteases, highlighting its high specificity and noting drawbacks such as off-target cleavage and sensitivity to chelators or reducing agents. Studies support enterokinase as a useful tool for producing tag-free proteins with native termini, provided optimal buffer conditions are maintained [156,157].

#### 4.1.4. HRV 3C and PreScission Protease

HRV 3C protease (PreScission™) recognizes the LEVLFQ↓GP cleavage motif and performs highly specific tag removal, leaving a minimal Gly–Pro dipeptide at the N-terminus. This 22 kDa protease operates efficiently at 4 °C, making it ideal for labile proteins and high-throughput workflows [158]. The molecular weight of PreScission Protease (Cytiva) is approximately 46 kDa. It has a GST (Glutathione S-transferase) tag attached to it for easier removal [159]. A comprehensive review [139] covered PreScission alongside TEV, thrombin, and others, emphasizing HRV 3C’s biochemical advantages, especially mild-temperature operation and buffer robustness, but also noting the residual GP dipeptide and occasionally slower cleavage rates compared to TEV protease in certain constructs.

#### 4.1.5. Tobacco Etch Virus (TEV) Protease

TEV protease is a cysteine protease from Tobacco Etch Virus that recognizes the ENLYFQ↓S/G consensus and cleaves with high sequence specificity, enabling precise tag removal with minimal residual amino acids (often only the natural N-termial residue) [160]. This is the most commonly used tag removal protease. The molecular weight of TEV protease is 27 KDa. Comprehensive enzymatic tag-removal reviews highlight TEV’s advantages over other proteases—including broad N-terminal residue tolerance, mild-temperature activity (including at 4 °C), and robust cleavage efficiency, while also pointing out limitations such as occasional autolysis (that can be mitigated by S219V mutation in the TEV enzyme), reduced activity at higher temperatures, and the need for reducing agents in some constructs [139,161,162].

### 4.2. Advances in Tagging Systems That Preserve Native Protein Sequences

In the case of conventional tags, an additional sequence is inevitably introduced at the linker region, which can interfere with accurate evaluation of the protein’s function. To address this issue, several tag systems have been developed that do not introduce extra N-terminal amino acids, and commercial kits are now available. Technical improvements in this area are still ongoing, as summarized in Table 4.

#### 4.2.1. eXact Tag and Profinity eXact Resin

The Profinity eXact™ tag system (Bio-Rad) enables rapid, tag-free recombinant protein purification using a self-cleaving N-terminal 75-amino-acid tag and immobilized subtilisin protease (S189 mutant derived from *Bacillus subtilis*) [163,164]. Cleavage is precisely triggered by fluoride ions, yielding native proteins with no residual tags.

Although a limitation is that halogenated compounds containing chloride ions (e.g., Tris-HCl, NaCl) cannot be used in the lysis or binding buffers, the system offers the advantage of one-step purification (~1 h) with high specificity, as has been reported in several applications involving recombinant protein expression and purification using this system [43,44,165,166]. Further optimization of the binding buffer conditions may enhance its performance.

#### 4.2.2. SUMO Tag and Protease

SUMO protease (*S. cerevisiae* Ulp1) targets the folded SUMO domain and cleaves precisely after the C-terminal GlyGly↓sequence of SUMO, releasing the native N-terminus with no extra residues—a clear advantage over peptide-cleaving proteases [72,167,168,169]. SUMO protease was sufficient to cleave mutated substrates with preferential activity of Gly > Ala > Ser > Cys at the P1 position and Gly = Ala = Ser = Cys at the P2 position [170]. A broader methodological review reports that SUMO proteases maintain high cleavage fidelity across diverse conditions, temperature (4–37 °C), pH (5.5–10.5), detergents, 2 M urea, tolerate various fusion partners, and rarely show off-target cleavage—though they cannot cleave proteins containing N-terminal proline immediately following the SUMO fusion [72,145].

#### 4.2.3. CASPON Tag and cpCasp2 Protease

The caspase-based fusion process (CASPON) tag system, developed by Boehringer Ingelheim, combines a solubility-enhancing tag (T7AC), a specific protease cleavage site (for circularly permuted caspase-2), and a hexa-histidine tag for affinity purification [171,172,173]. After purification via IMAC, the tag is efficiently removed by the cpCasp2 protease, leaving a native N-terminus. This system enables scalable, tag-free recombinant protein production under mild conditions.

#### 4.2.4. IMPACT

The IMPACT (Intein-Mediated Purification with an Affinity Chitin-binding Tag) system uses a chitin-binding domain (CBD) fused to an intein (a self-cleaving protein element) and the target protein [174,175]. After binding the fusion protein to a chitin resin via the CBD, the intein undergoes a controlled cleavage (usually triggered by thiol reagents like dithiothreitol), releasing the purified target protein with a native N- or C-terminus. This method allows gentle, tag-free purification without requiring external proteases [165,176,177].

### 4.3. Novel Tagging and Purification Strategies Using the Splint Intein System

In recent years, numerous studies have reported the development of novel tags and purification systems based on split inteins [178,179,180,181,182,183]. Beyond their use in recombinant protein purification, these systems have also led to the emergence of a new technology known as protein editing, cyclic peptide and protein generation, conditional protein splicing, protein labeling, bioconjugation, and synthetic biology.

As it is not feasible to cover all developments in this area, we will focus on the most representative split inteins.

#### 4.3.1. ProteinSelect Tag

The ProteinSelect tag system, developed by Cytiva, is a self-cleaving affinity tag based on a split intein derived from Ssp DnaE split intein [184,185,186,187]. The 36-amino-acid tag is fused to the N-terminus of the POI. Cleavage occurs spontaneously after binding to the resin, without the need for external inducers like pH shift or chemicals. The system enables one-step purification, tag cleavage, and elution, yielding tag-free proteins with over 98% purity. It is compatible with bacterial and mammalian expression systems and can be combined with other tags. However, cleavage efficiency is affected by the POI’s N-terminal residues, especially proline, which can inhibit cleavage. Resin is reusable for multiple cycles.

#### 4.3.2. iCap Tag

The iCap Tag system is a self-cleaving tag technology based on a modified Npu DnaE split intein, developed by Ohio State University and Protein Capture Science [188,189]. The N-terminal intein fragment is immobilized on resin, while the C-terminal fragment (36 amino acids) is fused to the POI. Cleavage is induced by lowering the pH (e.g., to 6.2), releasing the tag-free POI with high purity and no residual amino acids. A zinc-binding motif engineered into the intein enhances pH sensitivity and suppresses premature cleavage. The system avoids toxic chemicals, enables efficient purification, and supports resin regeneration. Cleavage efficiency depends on pH, temperature, and the N-terminal amino acid of the POI—proline at position one inhibits cleavage. Despite this, the system is highly controllable and suitable for diverse proteins in pharmaceutical applications.

#### 4.3.3. Gp41-1 Tag

The cyanophage-like gp41-1 tag is a highly active split intein system discovered through metagenomic analysis, showing exceptional potential for protein purification [190,191,192,193]. Derived from a gene associated with the gp41 DNA helicase, it exhibits protein trans-splicing activity and operates through pH-dependent cleavage. Compared to the widely used Npu DnaE intein, gp41-1 demonstrates up to 10 times faster splicing at physiological temperature, with high yields (85–95%) across a broad pH range (6–10) and temperature range. Structurally, gp41-1 is one of the smallest known inteins (125 amino acids total), with reversed charge distribution between its N- and C-intein domains compared to Npu DnaE. This compact size contributes to reduced steric hindrance and increased flexibility. The system has been effectively applied in affinity chromatography by immobilizing the N-intein on resin and attaching the C-intein to the POI. Cleavage is then induced by lowering the pH, allowing tag-free POI elution with high purity. While the C-terminal cleavage mutant is slower, it offers precise pH control. The gp41-1 system has been patented and shows promise as a universal, time- and cost-efficient platform for tag-free protein purification, although its sensitivity to mutations at extein junctions may limit its generalizability.

#### 4.3.4. Cfa DnaE Tag

The Cfa DnaE tag is an artificially engineered split intein based on a consensus sequence of Npu DnaE and Ssp DnaE inteins [184,188,193,194,195]. Designed for enhanced stability and fast splicing, it shows 2.5 times higher activity than Npu DnaE and tolerates harsh conditions, such as high temperatures and denaturants. It enables tag-free protein purification with high efficiency and no residual amino acids. Modifications, such as glycine insertions in the C-intein, improve elution and resin reuse. However, some variants rely on DTT for cleavage, which may limit use with disulfide-containing proteins in industrial settings.

#### 4.3.5. Npu DnaE

The Npu DnaE intein is a naturally split intein derived from the *DnaE* gene of the cyanobacterium *Nostoc punctiforme* (strain PCC 73102) [179,196]. It is one of the most widely used split inteins in protein engineering due to its outstanding performance in protein trans-splicing.

#### 4.3.6. Aes (Cysteine-Less Aes123 PolB1 Intein)

This is the most recently reported split intein as of 2025 [197]. Using structural, computational, and biochemical approaches, the authors engineered a monomeric, cysteine-less variant of the Aes123 PolB1 split intein—originally identified in T4-like bacteriophages infecting *Aeromonas salmonicida*. The engineered intein exhibits near-perfect splicing efficiency and ultrafast reaction kinetics. Their study also highlights precursor aggregation as a common challenge in split intein systems.

**Table 4 cimb-47-00768-t004:** Advances in tagging systems that preserve native protein sequences.

System	Tag	Size	Main Uses (with Fusion/Supplier/Resin/Cleavage Info)	Advantages	Disadvantages	Ref.
Protease-based system	eXact-tag	8 kDa	N-terminal fusion; Bio-Rad Laboratories; Profinity eXact resin; cleavage by NaF (requires halogen-free buffers) → Enhancing solubility and purification of expressed proteins	Stable expression in *E. coli*; self-cleavage leaves no extra residues	Requires specific cleavage conditions; commercial system dependent; binding and washing require halogen-free buffers (no Cl^−^, etc.)	[163,164]
	SUMO-tag	12 kDa	N-terminal fusion; Merck KGaA; SUMO-tag TRAP or His/GST resin (if fused); cleavage by SUMO protease (Ulp1) → Solubilization, increased expression, purification	Strong solubilization effect; SUMO protease provides precise cleavage without residual amino acids	Protease cost is high; may interfere with endogenous SUMO in eukaryotic systems	[167,168,169]
	CASPON-tag	4.1 kDa	N-terminal fusion; Boehringer Ingelheim (patent); IMAC capture; cleavage by Caspase → Purification and high-purity protein production	Residue-free cleavage; compatible with His-tag purification	Dependent on CASPON protease; requires proprietary technology	[171,172,173]
Intein-based system	IMPACT (Intein-Mediated Purification with an Affinity Chitin-binding Tag)	27 kDa (CBD-tag)	N- or C-terminal fusion; New England BioLabs; chitin resin; cleavage by reducing reagent (DTT) → Purification of insoluble proteins (intein + CBD system)	Intein-mediated self-cleavage allows tag removal; suitable for large-scale purification	Cleavage efficiency depends on optimized conditions; incomplete cleavage possible	[174,175]
Split intein-based system	Protein select (modified Ssp DnaE)	4 kDa	N-terminal fusion; Cytiva; Protein Select resin; natural on-column self-cleavage → Affinity capture and traceless purification	Self-cleaving tag enabling one-step capture and traceless release (no protease required); simplifies workflows	Proprietary system requiring specialized resin; tag cleavage is not time-controllable (autocleavage occurs spontaneously upon folding/binding); limited to N-terminal fusion; efficiency can vary with target folding	[184,185,186,187]
	iCap	4 kDa	N-terminal fusion; Protein Capture Science (Ohio Univ. tech); NpuN immobilized resin; cleavage triggered by pH shift (8.5 → 6.2) → Precise cleavage for native protein recovery	Protease-mediated exact cleavage without residual residues	Limited applicability; strongly condition dependent	[188,189]
	Modified Npu DNaE	17.2 kDa	N-terminal fusion; Merck KGaA; NpuN immobilized resin; cleavage triggered by pH shift → Split intein-based protein ligation/editing	High-efficiency splicing and self-cleavage; seamless ligation	Complex system design; requires optimization in vitro	[179,196]
	Gp41-1	24.2 kDa	N- or C-terminal fusion; immobilized Gp41-1N resin; cleavage triggered by pH shift (9 → 7) → Split intein-mediated protein ligation/editing	Highly efficient; functions under broad conditions	Increases protein size; requires careful design	[190,191,192,193]
	Cfa	17.2 kDa	C-terminal fusion; IMAC or chitin resin; cleavage triggered by reassociation of N/C intein fragments → Split intein-mediated protein ligation	Efficient, residue-free editing	Strongly condition-dependent; complex recombination construction	[194,195]

## 5. Conclusions

Fusion tags are indispensable for recombinant protein expression, helping to overcome insolubility, low yields, and instability. Protein tags such as GFP, Trx, MBP, NusA, SUMO, GST, and halophilic β-lactamase improve solubility and folding, while carriers like HSA and Fc domains extend serum half-life and therapeutic utility. Small peptide tags, including His, Flag, Myc, PA, HA, V5, and Strep, are widely used for purification and detection, with the His-tag being the most practical and cost-effective option. Advances in tag removal, particularly split intein technologies, have enabled seamless production of tag-free proteins and opened a completely new field of protein editing. Emerging trends, including de novo designed tags and tags for specific delivery, further expand the versatility of fusion tags and hold transformative potential for protein engineering and therapeutics.

## Figures and Tables

**Figure 1 cimb-47-00768-f001:**
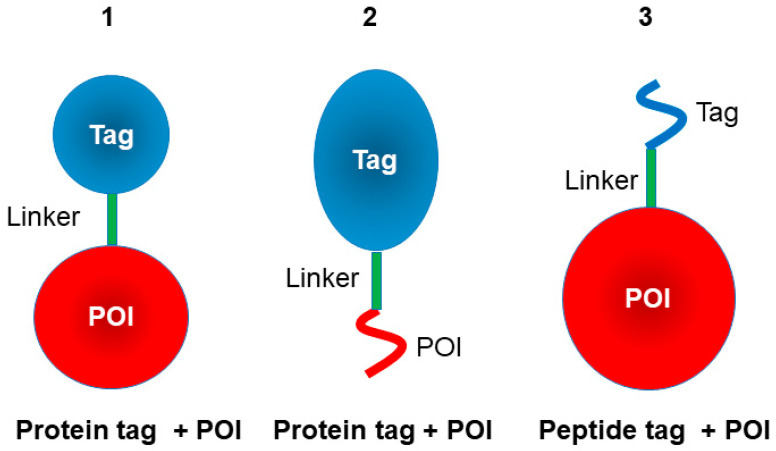
Fusion-tag construction. Tags can be positioned at either the N- or C-terminus of the protein. When placed at the N-terminus, the tag’s C-terminus is fused to the N-terminus of the linker, which in turn connects via its C-terminus to the N-terminus of the POI. Conversely, when positioned at the C-terminus, the tag’s N-terminus is fused to the C-terminus of the linker, which is then fused to the C-terminus of the POI. Color scheme: Tag: pale blue; Linker: green; POI: red.

**Figure 2 cimb-47-00768-f002:**
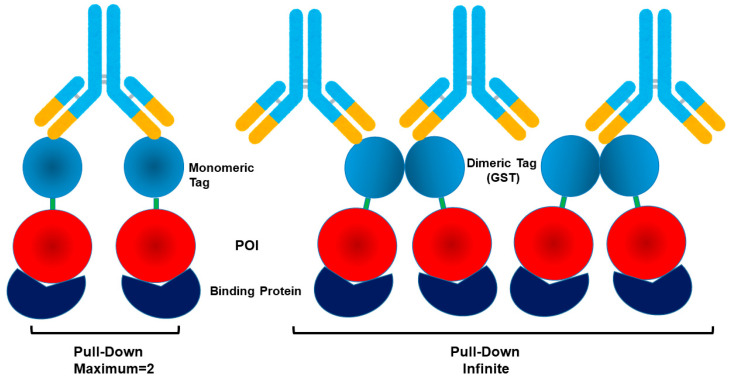
Binding modes of monomeric and multimeric fusion tags in pull-down assay. Antibody-mediated pull-down leads to binding of two proteins for a monomeric fusion tag and a potential infinite number of proteins for multimeric fusion tag. Color scheme: Antibody—constant region, light blue; variable region, yellow; Tag, pale blue; POI, red; Binding protein, dark blue.

**Figure 3 cimb-47-00768-f003:**
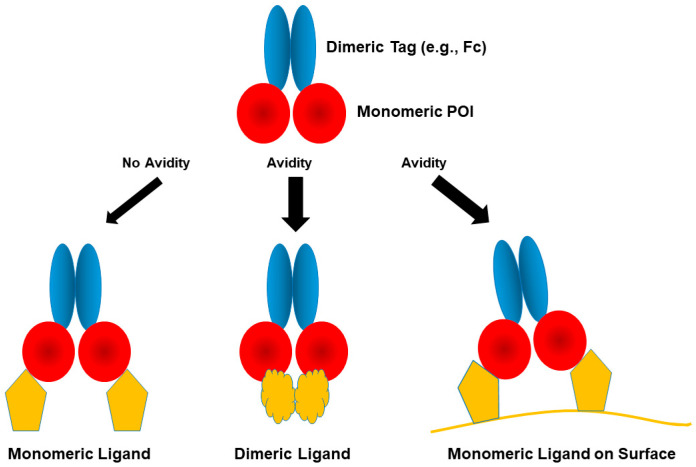
Binding avidity of dimeric fusion tags.

**Figure 4 cimb-47-00768-f004:**
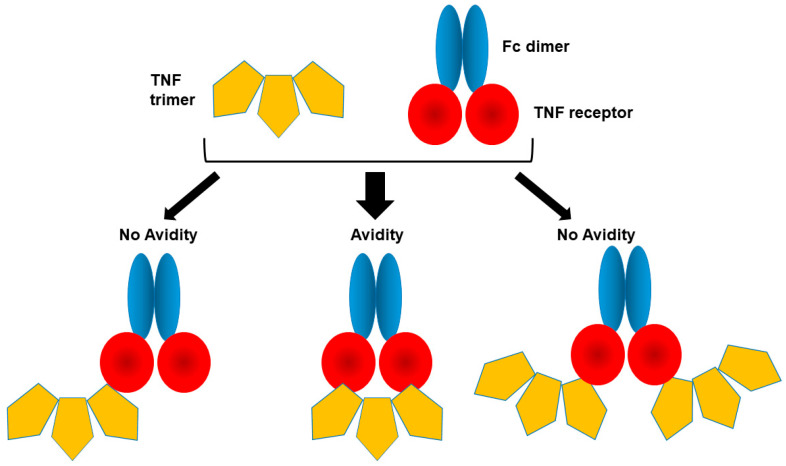
Binding models of etanercept to TNF trimer.

**Figure 5 cimb-47-00768-f005:**
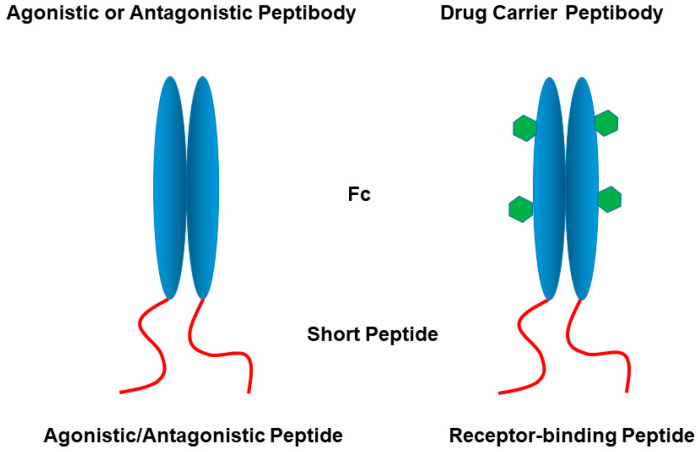
Schematic illustration of peptibodies. Green hexagons indicate conjugated drugs.

**Figure 6 cimb-47-00768-f006:**
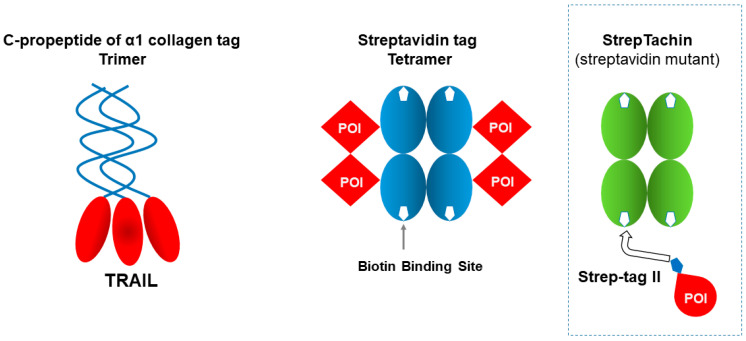
Structural models of trimeric and tetrameric tags.

**Figure 7 cimb-47-00768-f007:**
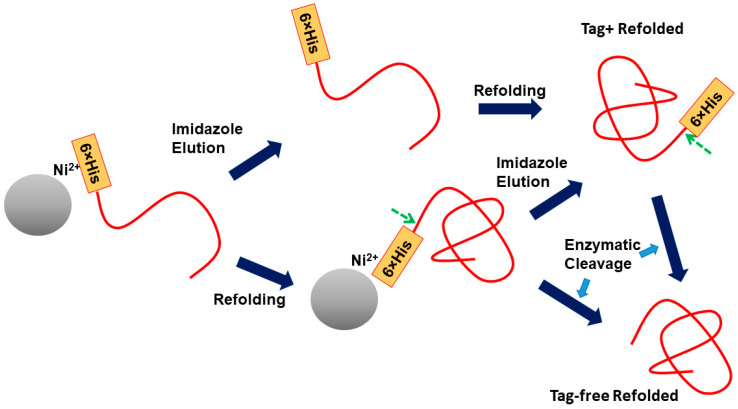
Schematic illustration of His-tag proteins in the presence of denaturant and subsequent refolding. Green arrows indicate cleavage sites.

**Figure 8 cimb-47-00768-f008:**
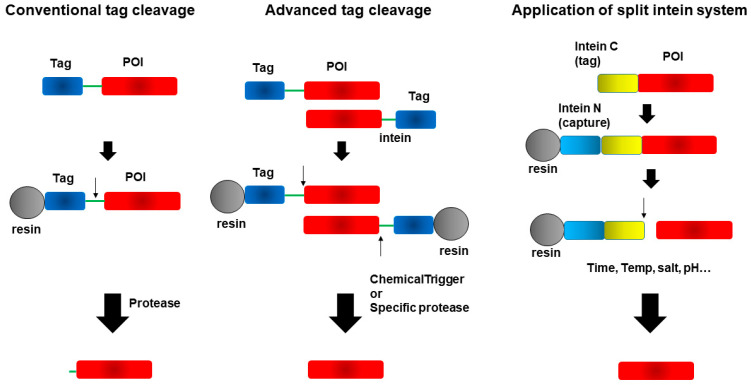
Cleavage of fusion tags from target proteins: Evolution of tag removal strategies. The ‘↓’ symbol indicates the cleavage site.

## Data Availability

No new data were created or analyzed in this study. Data sharing is not applicable to this article.

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
