# Peer review of "Beyond Purification: Evolving Roles of Fusion Tags in Biotechnology"

_cimb, 2025, doi:10.3390/cimb47090768_

Round 1

Reviewer 1 Report

Comments and Suggestions for Authors

In general, the review is well written. For some of the tags plenty of examples are given, and some are mentioned very briefly. This a review, so the reader expects to get as much information as possible. This information must be well synthesized, which the authors managed to do in some of the sections.

In sections 2.7 it was mentioned several times that the fusion of a His-tag does not increase the solubility of the POI. The main function of His -tag, however, is to facilitate the purification, not to increase the solubility. The authors should consider that.

I think it is a good idea to include, very briefly, the pros and cons of each mentioned tag at the end of each section.

Line 35: A number of protein or peptide-based fusion tags has been developed and below is some of them, although there may be more. – This sentence, which is grammatically incorrect, must be modified. There are extensive number of literature reviews on the topic of fusion proteins, which can be cited instead of saying “although there may be more”.

Fig. 1: It is better to address the position of the tag as either C-terminal or N-terminal. The title of the figure should be changed.

Line 46:  27-30 kDa

Line 48-49: When GFP fused to a protein of interest (POI) or “target protein” as GFP-POI or POI-GFP and expressed… - When GFP fused to a protein of interest (POI) or “target protein” as N- or C-terminal tag …

Line 84: They expressed 8 human proteins – Who is “they”?

Line 91: soluble when expressed in E. coli, made soluble by fusing to Trx tag … and line 92- of a cytokine, interleukin-24, was made using Trx-fusion expression… - these sentences are not grammatically correct.

Line 92: It was fused – Written that way it sounds like IL-24 was used as fusion tag, not Trx.

Line 127: HSA fusion with human serum albumin – the citation is not correct. Human lactoferrin was fused to HSA.

Line 135: SUMO was found effective in increasing soluble expression of many proteins, including SARS-CoV-2 – May be the authors meant “including some of the SARS-CoV-2 proteins”?

Section 2.9 : The first and second paragraphs should be rewritten for clarity.

Line 274-281: I don’t think that this information should be included in a review paper.

Line 294-295: His-tag does not bind to a chromatography. It binds to a resin!

Section 3.6: If the authors want to include the sequences of each peptide tag, they should include the aa sequence of V5.

Line 411: I suppose that Figure 7 is typo.

Line 419 - Supplier literature?

Reviewer 2 Report

Comments and Suggestions for Authors

Add some more specific sentences about evolving roles of tags in the abstract.

Introduction should contain more information about the advent of these tags to set a background for the explanation later in the article.

Spelling corrections: I.e., dimerization in table,

Please provide references in all the tables, some limitations are similar for protein and peptide tags i.e., dimerization, insoluble, GST and Fc in table 1 and 2.

Figure 7 is missing,

Only little details have been added in the text under headings 2.3, 2.4, 2.6, 2.10, 2.11, 3.2, 3.3, 3.4, 3.5, 3.6, 3.7, 3.8. More details should be included.

Main contents of the tables should be same such as advantages, limitations, size, use etc in all the tables.

A brief conclusion at the end of each tag would improve the quality of the article.

The information under heading 4.1 is missing. Add few sentences for each protease on its most common application or a key advantage/disadvantage.

Please add emerging trends in the conclusion in addition to inteins. For example, de novo designed tags and tags for specific delivery.

Line 29 replace usion tags with fusion tags.

Line 70 remove full stop.

Comments on the Quality of English Language

The manuscript as a whole needs to be revised in order to eliminate grammatical errors.

Round 2

Reviewer 1 Report

Comments and Suggestions for Authors

I appreciate the authors’ efforts to address all the remarks – the manuscript is now considerably improved.

Reviewer 2 Report

Comments and Suggestions for Authors

Please remove the sentence last sentence from all the tags stating "In conclusion" to avoid redundancy. It is unnecessary, or rephrase it in all the tags in different ways.

Please re-draw the figure 1 to align Tag, Linker and POI with their respective parts, blue, green and red.  
